# Exploring Weight Stigma in Saudi Arabia: A Nationwide Cross-Sectional Study

**DOI:** 10.3390/ijerph18179141

**Published:** 2021-08-30

**Authors:** Nora A. Althumiri, Mada H. Basyouni, Norah AlMousa, Mohammed F. AlJuwaysim, Adel A. Alhamdan, Faisal Saeed Al-Qahtani, Nasser F. BinDhim, Saleh A. Alqahtani

**Affiliations:** 1Sharik Association for Health Research, Riyadh 13326, Saudi Arabia; mada.basyouni@sharikhealth.net (M.H.B.); 2170001797@iau.edu.sa (N.A.); 216006361@student.kfu.edu.sa (M.F.A.); nasser.bindhim@sharikhealth.net (N.F.B.); 2Ministry of Health, Riyadh 11176, Saudi Arabia; 3Public Health College, Imam Abdulrahman Bin Faisal University, Dammam 31441, Saudi Arabia; 4Pharmacy College, King Faisal University, AlAhsa 31982, Saudi Arabia; 5Community Health Sciences, College of Applied Medical Sciences, King Saud University, Riyadh 11362, Saudi Arabia; adel@ksu.edu.sa; 6Department of Family & Community Medicine, College of Medicine, King Khalid University, Abha 61421, Saudi Arabia; drfaisalqahtani@gmail.com; 7College of Medicine, Alfaisal University, Riyadh 11533, Saudi Arabia; 8Saudi Food and Drug Authority, Riyadh 13513, Saudi Arabia; 9Liver Transplant Unit, King Faisal Specialist Hospital & Research Centre, Riyadh 11211, Saudi Arabia; salqaht1@jhmi.edu; 10Division of Gastroenterology and Hepatology, Johns Hopkins University, Baltimore, MD 21218, USA

**Keywords:** weight stigma, obesity, Saudi Arabia, risk factors

## Abstract

Background: Weight stigma (WS) in the Middle East, especially in Saudi Arabia, is widely ignored. People with obesity are blamed for their weight, and there is a common perception that weight stigmatization is justifiable and may motivate individuals to adopt healthier behaviors. The authors of this study aimed to explore WS prevalence and factors associated with WS in a large nationwide study of Saudi Arabian adults. Methods: This study was a nationwide cross-sectional survey conducted via phone interviews in June 2020. A proportional quota-sampling technique was adopted to obtain equal distributions of participants by age and sex across the 13 regions of Saudi Arabia. In total, 6239 people were contacted, and 4709 (75.48%) responded and completed the interview. The authors of the study collected data about WS using the Arabic Weight Self-Stigma Questionnaire (WSSQ), BMI, smoking, nutritional knowledge, bariatric surgery, risk of depression, and demographic variables. Results: Participants had a mean age of 36.4 ± 13.5 (18–90), and 50.1% were female. The prevalence of higher WS was 46.4%. Among other risk factors, there was a significant association between WS and obesity (odds ratio (OR): 3.93; 95% CI: 2.83–5.44; *p* < 0.001), waterpipe smoking (OR: 1.80; 95% CI: 1.20–2.69; *p* < 0.001), bariatric surgery (OR: 2.07; 95% CI: 1.53–2.81; *p* < 0.001), and risk of depression (OR: 1.68; 95% CI: 1.36–2.09; *p* < 0.001). Conclusion: This was the first study to explore WS and its associated factors among adults in a community setting in Saudi Arabia. This study revealed some risk factors associated with WS that may help to identify people at risk of WS and to develop interventions to reduce WS, such as improving nutritional knowledge, correcting the ideas about bariatric surgery and obesity in general, and ceasing waterpipe smoking.

## 1. Introduction

Weight stigma (WS) (also known as weight bias or weight discrimination) is defined as “discrimination or stereotyping based on a person’s weight” [1]. Research in many countries has demonstrated that people have negative attitudes toward persons with overweight or obesity [2]. Numerous studies have documented harmful weight-related stereotypes, such as ideas that people with overweight or obesity are lazy, weak-willed, unsuccessful, and unintelligent; have a lack self-discipline; and are noncompliant with weight-loss treatments [3,4].

WS can have a negative effect on multiple domains of living, both internally—through interpersonal relationships, greater body shame, and lower self-compassion, all of which lead to greater psychological distress, higher perceived loneliness, lower satisfaction with life, and a higher risk of developing eating disorders [5]—and externally—through such things as employment, education, health care, and mass media [3,5]. In addition, many studies have found associations between weight stigma and internalized weight stigma in a wide range of problematic eating behaviors that occur for both adults and children, even after controlling for many factors, such as body mass index (BMI), self-esteem, mood disorders, and other potential confounders [6]. Unfortunately, WS is still a socially acceptable form of stigma that often occurs and is tolerated due to beliefs that stigma and shame will motivate people to lose weight [7]. WS has been frequently reported by people in various social and professional groups including employers, coworkers, teachers, physicians, nurses, medical students, dietitians, psychologists, peers, friends, family members, and even children as young as three years old [2,8].

With the prevalence of obesity in Saudi Arabia growing and estimated to increase to 41% of men and 78% of women by 2022, the importance of curbing WS is evident [9]. However, there is a lack of information on WS in terms of prevalence. Only a few articles about “weight stigma prevalence”, “weight bias prevalence”, and “weight discrimination prevalence” have been published, and none of them have concerned WS in Saudi Arabia. However, exposure to mass media has been associated with changing the idea of a perfect body shape in various Arab region countries [10,11,12]. Furthermore, a recent study in Saudi Arabia showed that 42% of participants did not have an accurate perception of their weight; 67.6% of obese participants misclassified their weight, compared to 33.9% in normal weight participants [13]. Another study in 2014 revealed that Saudi women believe that obesity attracts stigma and morally compromising activities [14].

One large online study of adults in a commercial weight management program in the United States showed that internalized WS was more prevalent than in the general population and higher among participants who were female, younger, and had higher BMI (*p* < 0.001) [15]. Another cross-sectional study based in the United States conducted with >3800 adults who completed an online survey showed that the prevalence of WS was 57% [8]. Additionally, they found that the odds of internalized WS were higher among people with overweight or obesity and those who believe individuals are personally responsible for their body weight [8].

The Weight Self-Stigma Questionnaire (WSSQ) is a survey that assesses internalized weight stigma and has been used globally; however, its application in Arabic-speaking countries has been limited due to language barriers [5]. With the recent translation and validation of an Arabic version of the WSSQ, the opportunity to apply it to a Saudi Arabian population is now available [5]. Thus, we aimed to explore WS prevalence and factors associated with WS using the Arabic WSSQ in a large nationwide study of Saudi Arabian adults.

## 2. Materials and Methods

### 2.1. Study Design

This study was a cross-sectional survey (Sharik Diet and Health National Survey) [16] with national coverage that was conducted in the Kingdom of Saudi Arabia via computer-assisted [17] phone interviews in June 2020. Previous studies have used the same dataset [18,19].

### 2.2. Sampling and Sample Size

A proportional quota sampling technique was used to generate an equal distribution of participants based on three variables (age, sex, and region). The regions included the 13 administrative regions of Saudi Arabia (Aljouf, Northern Borders, Tabuk, Hail, Madinah, Qassim, Makkah, Riyadh, Eastern Region, Baha, Asir, Jazan, and Najran). Figure 1 shows a map of Saudi Arabia, highlighting its regions and their adult population proportions. We used two age groups, which were established based on the median age of Saudi Arabian adults (36) (age group 1: 18–36; age group 2: 37 and above). Thus, 52 sampling quotas were generated. The required sample size was calculated based on a medium effect size of approximately 0.25, with 80% power and 95% CI, to compare age and sex across regions [20]. Ninety participants were required to meet each quota, and the total targeted sample size calculated for this study was 4680 participants.

The Z-DataCloud^®^ research data collection system, which has integrated eligibility testing and sampling control tools and algorithms, was used to control the sample distribution and eliminate sampling bias [17]. The eligibility testing tool included three sampling variables to automatically determine adherence to the sampling quotas, including those based on age, sex, and region.

### 2.3. Participant Recruitment

Participant recruitment was limited to the Arabic-speaking population (more than 80% of the population), adults (≥18 years old), and residents of Saudi Arabia. A random mobile phone number list was generated from the Sharik Association for Research and Studies to identify potential participants [21]. The Sharik database comprises individuals willing to participate in research projects that have consented to be contacted for future studies. The database contains more than 76 thousand individuals distributed across the 13 administrative regions of Saudi Arabia and continues to grow [21]. Individuals were contacted by phone on up to three occasions. If there was no response, another individual generated from the same database with identical sampling variables was contacted. After individuals consented to participate, the interviewer assessed the individuals’ eligibility with Z-DataCloud^®^ based on the abovementioned quota completion criteria. Once the quota was complete, it was automatically closed.

### 2.4. Survey and Outcome Measures

Each interview lasted approximately 8 min and was conducted by a trained data collector. Demographic information (age, sex, education level, and region), and information about WS, health, and lifestyle habits, was captured.

We used the Arabic translation of the WSSQ, which was validated in a Saudi Arabian population [5]. The WSSQ is a 12-item Likert-type measure of weight-related self-stigmatization [5]. The items in the WSSQ are rated on a scale from 1 (completely disagree) to 5 (completely agree) [5]. The tool was translated using standard backward and forward translation and two focus groups from Saudi Arabia that were asked to answer and discuss the questionnaire; their comments and understanding of each item’s meaning were discussed and taken into consideration in the final version [5]. As there is no cutoff point for the WSSQ, we categorized the scores into two categories (0 = the overall median score or lower “lower WS” and 1 = above the median score “higher WS”).

Basic nutritional knowledge was assessed by asking the participants 4 true-or-false statements developed for this study: (1) “Sugar should be the main source of calories”. (2) “The portion size is the total number of calories that come from carbohydrates and protein”. (3) “Protein is high in sugar, which is why it is recommended to lower your intake of it”. (4) “To lose weight, generally it recommended to consume more than three portions of fruits every day”. Then, the score was tabulated from 0 to 4 based on how many correct answers the participants could provide.

Self-reported weight and height were collected, and then BMI was calculated; to increase the quality of the self-reported weight, we asked the participants when they last measured their weight. BMI was categorized into four groups: underweight (<18.5 kg/m^2^), normal weight (18.5–24.9 kg/m^2^), overweight (25–29.9 kg/m^2^), and obese (≥30 kg/m^2^) according to the categories used by the Centers for Disease Control and Prevention (CDC) [22]. Weight misclassification was assessed using the NHANES 2017–2018 weight history questionnaire by asking the participant, “Do you consider yourself to be overweight, obese, underweight or about the right weight?” Then, the participants’ answers were linked with their BMI. If the participant’s classification for their weight did not match the BMI category, then the weight misclassification status variable was “yes”; if the answer matched the BMI category, the weight misclassification status was “no” [23].

Overall health was assessed using a question from the World Health Organization (WHO) World Health Survey (“In general, how do you rate your overall health?”; possible answers were “great”, “very good”, “good”, “normal”, and “bad”) [24]. Participants were asked about their sleeping patterns (“Do you have trouble sleeping?”). We also asked the participants whether they had received doctor advice about their weight in the last 6 months.

The risk of depression was assessed using PHQ-9 [25,26,27], in which a score above ten denotes a high risk of depression [28]. The PHQ-9 has been used for mental health screening in many local and international surveys and surveillance systems (e.g., the CDC in the United States uses the PHQ-9 in the Behavioral Risk Factor Surveillance System and the National Health and Nutrition Examination Survey, and it is used in Saudi Arabia’s national mental health surveillance system), thereby allowing for international comparison [29,30].

Cigarette and waterpipe smoking were assessed by asking participants if they were current smokers with the possible answers of “daily smoker”, “occasional smoker”, or “nonsmoker”. The number of leisure sitting hours that participants reported were used to quantify sedentary behavior.

After the first draft of the survey was finalized, a linguistic validation to ensure the clarity and understanding of questions was conducted via a focus group comprising seven participants who were asked to discuss and answer the survey. According to the results of the focus group and feedback from the researchers and interviewers, the questionnaire was further edited, and a final version was produced.

## 3. Statistical Analysis

Population prevalence data were weighted to represent the adult population in Saudi Arabia according to the General Authority of Statistics Census Report [31]. Quantitative variables are presented as means and SDs if they had a normal distribution, or as medians and ranges, as appropriate. Qualitative variables are presented as percentages and confidence intervals (CIs) and were compared using Pearson’s χ^2^ test. A forward maximum likelihood logistic regression model, including all the demographic variables, lifestyle factors, nutritional knowledge, weight management actions, and BMI categories, was used to identify variables that are currently associated with higher WS. The results are reported according to the Strengthening the Reporting of Observational Studies in Epidemiology (STROBE) checklist for cross-sectional studies [32].

## 4. Results

### 4.1. Demographics and Response Rate

Out of the 6239 contacted individuals, 4709 (75.48%) agreed to participate and completed an interview. There was an equal distribution of participants among the main regions of Saudi Arabia. From the total sample, 50.1% were female, and the mean age was 36.4 (SD: 13.5). The median age was 36 (range: 18–90). The majority of the participants (59.6%) had a bachelor’s degree. Table 1 shows the demographic characteristics of the participants. The majority (70.1%) of participants indicated having weighed themselves within the past 30 days.

### 4.2. Prevalence of WS

The median WS score of all participants was 12. Overall, the national weighted prevalence of higher WS (cutoff above 12) was 46.4%. Participants aged 40–49 were the most affected participants, and Hail was the region with the lowest number of participants with higher WS. Table 2 shows the prevalence of WS by participant demographic characteristics.

### 4.3. Associations among WS, Lifestyle, and Basic Nutritional Knowledge

The majority of participants with bad overall health were in the higher WS category (62.2%). Table 3 shows the associations among WS, lifestyle, and basic nutritional knowledge.

### 4.4. Associations between WS and Current Weight Management-Related Variables

Of the obese participants, 68.4% were in the higher WS category. Table 4 shows the associations between WS and weight management, practice, and perception.

### 4.5. Factors Associated with Higher Weight Stigma

The regression model revealed that all variables included in the model were associated with WS except for education level, sex, and cigarette smoking. Obesity, bariatric surgery, and daily waterpipe smoking were the strongest factors associated with WS. Table 5 shows the regression model with all remaining variables.

## 5. Discussion

### 5.1. Results Summary

This cross-sectional study has shown the prevalence of weight stigma and associated risk factors in a nationwide sample of adults in Saudi Arabia. The results showed that the national weighted prevalence of higher WS was relatively high, at 46.4%. Demographic characteristics such as age and region were significantly associated with higher WS. This study has revealed some risk factors associated with WS. Obesity, bariatric surgery, and daily waterpipe smoking were the factors found to be most strongly associated with WS.

### 5.2. WS Prevalence

To our knowledge, this is the first national study in the Middle East to investigate the prevalence of WS and associated risk factors. However, globally, there is a lack of information and studies on the prevalence of WS, and very few studies have used validated tools to assess it. In addition, most prior studies have used an online self-reported questionnaire targeting specific groups or settings (such as people with obesity, individuals, or healthcare workers) [2]. Our findings showed that the prevalence of WS in Saudi Arabia is relatively high, at 46.4%. These results were generally similar to those of studies conducted in other countries [8,15]. A study conducted in U.S. adults showed that at least 44% of adults across samples had above-average WS [33]. Another cross-sectional study performed on 3800 participants via an online survey exploring participants’ encounters with weight-related intolerance, teasing, and beliefs regarding obesity showed that the prevalence of weight stigma in this sample was 57% [8].

### 5.3. WS and Obesity

Our results indicate a significant association between participants with higher BMI and WS. This was an expected finding, given that people with overweight or obesity are more likely to face weight discrimination. Previous studies have supported this finding [22,33,34,35], which might be linked to the fact that people are influenced by media and social influencers with perfect bodies [36]. It has been suggested that people compare their appearances to those of people on Instagram and other platforms, and they often judge themselves to be worse off [37]. Body image in contemporary societies is in general driven by mass media, the beauty business, and old views of health and wellbeing that define certain human body shapes and sizes as targets for monitoring and control [36,38]. The effect of global media on body image and the self-perception that obesity is associated with stigma are present issues in Saudi Arabia, and in other countries [10,11,12,14]. This evidence shows that WS is a cross-cultural issue.

### 5.4. WS and Bariatric Surgery

This study has shown a positive association between those who have a higher WS and individuals who have undergone bariatric surgery. This might be linked to the existing bariatric surgery stigma, in addition to the weight stigma. Psychological factors also play roles in evaluating WS, even after undergoing bariatric surgery. Recent evidence suggests that people who lose weight via bariatric surgery are more negatively appraised than are people who lose weight by changing their behavior and lifestyle, such as dieting and exercising [39,40]. The negative assessments of those who lose weight via surgical procedures might be a reason for the misperception of bariatric surgery as “an easy way out”, and a lack of effort from the individual when losing weight through bariatric surgery. Another study showed that participants rated individuals who lost weight through surgery as significantly lazier, sloppier, less competent, sociable, less attractive, and less healthy regarding eating habits [40]. In contrast, individuals who lost weight through diet and exercise were not evaluated as harshly [40]. Despite these misconceptions, people who undertake bariatric surgery to reduce their weight must keep a strict diet and exercise routine to stimulate weight loss and avoid weight regain later [41]. Thus, educating lay people about the efforts that bariatric surgery patients invest on their weight reduction journey might reduce some of the negative perceptions and stigma about the bariatric surgery and its patients [35,42].

### 5.5. WS and Waterpipe Smoking

We found a strong association between WS and the use of water pipes. Using and preparing a water pipe, which is typically an event lasting 45 min to an hour, might be one reason for a highly sedentary lifestyle that could keep individuals away from socialization [43]. There might indeed be an overlap, as both variables showed significant associations with WS in our regression model. Some studies have investigated and found a relationship between smoking water pipes and BMI [44]. One study indicated that daily waterpipe users had higher BMIs, translating into six extra kilograms of weight on average, and were three times as likely to have obesity [44,45]. However, to our knowledge, no studies have investigated the relationship between smoking waterpipes and WS.

### 5.6. WS and Nutrition Knowledge

In this study, we found that basic nutritional knowledge was associated with less WS. The regression model showed that this relationship was independent of the participant’s educational level. Though we were not able to identify a similar finding in the literature, basic nutrition knowledge is associated with various positive health outcomes, such as healthy eating [46]. Thus, improving nutritional knowledge could be a tool to assist in reducing the burden of WS. However, this effect needs to be further investigated for confirmation.

This study had some strengths and limitations. First, the sampling was strengthened through the use of 52 quota, potentially limiting the selection bias and generating a balanced research sample in terms of regions, sexes, and ages in Saudi Arabia. The large nation-wide sample size was a strength, especially when considering the scarce knowledge available on internalized weight stigma’s prevalence in the Saudi population. However, the use of sharik research participants’ database might also have introduced some sampling bias, given that registration in the database was voluntary. Nonetheless, such a research database is useful for recruiting large research samples without encountering restricting barriers. This type of recruitment was advantageous, considering that this project was executed during the COVID-19 pandemic, which could have otherwise hindered the recruitment process. Another limitation of this study was that we used the median instead of a fixed a cut-off points to classify WS, which might limit comparisons with future studies that use different cut-off points. Data integrity assessments, inherent to the Z-DataCloud^®^ research data collection system, minimized inaccurate data recording. Linguistic validation and survey testing were utilized to improve the questionnaire’s reliability.

## 6. Conclusions

This was the first study to explore WS and its associated factors among adults in a community setting in Saudi Arabia. This information will be valuable for clinicians and policymakers, particularly in terms of obesity and mental health. This study revealed some risk factors associated with WS that may help to identify people at risk of WS and to develop interventions to reduce WS, such as improving nutritional knowledge, correcting ideas about the bariatric surgery and obesity in general, and ceasing waterpipe smoking. This study’s results will also open the door for future research on WS, weight management, and characteristics in Saudi Arabia.

## Figures and Tables

**Figure 1 ijerph-18-09141-f001:**
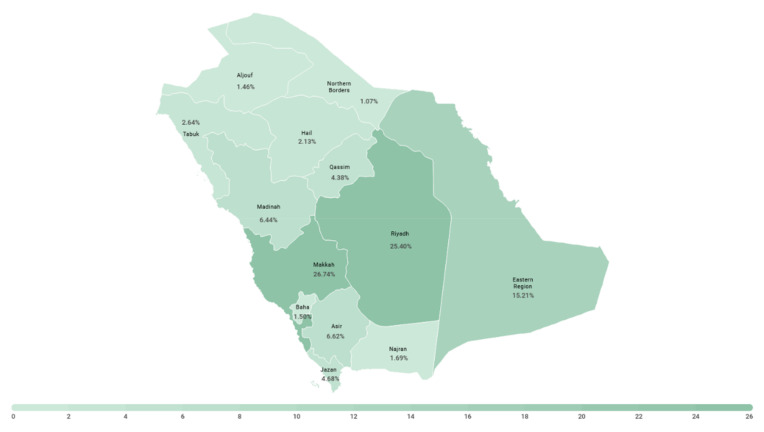
Map of the Kingdom of Saudi Arabia showing the distribution of the adult population.

**Table 1 ijerph-18-09141-t001:** Demographic characteristics of the participants (*n* = 4709).

Demographic Characteristics	Female *n* (%)	Male *n* (%)	Total *n* (%)
Age Group			
18–19	120 (47.1)	135 (52.9)	255 (5.4)
20–29	827 (53.1)	729 (46.9)	1556 (33.0)
30–39	469 (46.5)	540 (53.5)	1009 (21.4)
40–49	577 (55.3)	467 (44.7)	1044 (22.2)
50–59	260 (46.8)	295 (53.2)	555 (11.8)
60+	105 (36.2)	185 (63.8)	290 (6.2)
Region			
Aljouf	180 (49.9)	181 (50.1)	361 (7.7)
Northern Borders	181 (50.1)	180 (49.9)	361 (7.7)
Tabuk	182 (50.3)	180 (49.7)	362 (7.7)
Hail	179 (49.9)	180 (50.1)	359 (7.6)
Madinah	184 (50.3)	182 (49.7)	366 (7.8)
Qassim	181 (50.1)	180 (49.9)	361 (7.7)
Makkah	181 (50.3)	181 (50.0)	362 (7.7)
Riyadh	183 (50.3)	181 (49.7)	364 (7.7)
Eastern Region	180 (50.0)	180 (50.0)	360 (7.6)
Baha	181 (49.7)	183 (50.3)	364 (7.7)
Asir	184 (50.3)	182 (49.7)	366 (7.8)
Jazan	181 (50.1)	180 (49.9)	361 (7.7)
Najran	181 (50.0)	181 (50.0)	362 (7.7)
Sex			
Female	-	-	2358 (50.1)
Male	-	-	2351 (49.9)
Educational Level			
High School or Less	894 (49.9)	899 (50.1)	1973 (38.1)
Undergraduate Diploma	238 (41.2)	339 (58.8)	577 (12.3)
Bachelor’s Degree	1145 (54.5)	957 (45.5)	2102 (44.6)
Postgraduate Degree (Master’s/PhD)	80 (33.9)	156 (66.1)	236 (5.0)

**Table 2 ijerph-18-09141-t002:** Prevalence of weight stigma by participant demographic characteristics (*n* = 4709).

Variable	Lower WS * *n* (%)	Higher WS *n* (%)	Chi-Square *p*-Value
Sex			
Male	1351 (57.5)	1000 (42.5)	<0.001
Female	1174 (49.8)	1184 (50.2)	
Age Group			
18–19	135 (52.9)	120 (47.1)	0.023
20–29	859 (55.2)	697 (44.8)	
30–39	554 (54.9)	455 (45.1)	
40–49	510 (48.9)	534 (51.1)	
50–59	303 (54.6)	252 (45.4)	
60+	164 (56.6)	126 (43.4)	
Region			
Jouf	199 (55.1)	162 (44.9)	<0.001
Northern Border	165 (45.7)	196 (54.3)	
Tabuk	193 (53.3)	169 (46.7)	
Hail	223 (62.1)	136 (37.0)	
Madinah	206 (56.3)	160 (43.7)	
Qassim	213 (59.0)	148 (41.0)	
Riyadh	185 (3.9)	179 (49.2)	
Eastern Region	196 (54.4)	164 (45.6)	
Baha	205 (56.3)	159 (43.7)	
Asir	181 (49.5)	185 (50.5)	
Jazan	212 (58.7)	149 (41.3)	
Najran	160 (44.2)	202 (55.8)	
Education Level			
High School or Less	966 (53.9)	827 (46.1)	0.591
Undergraduate Diploma	308 (53.4)	269 (46.6)	
Bachelor’s Degree	1114 (53.0)	988 (47.0)	
Postgraduate Degree (Master’s/PhD)	136 (57.6)	100 (42.4)	

* (Lower WS) median score or lower weight stigma. (Higher WS) above median score weight stigma.

**Table 3 ijerph-18-09141-t003:** Associations among WS, lifestyle, and basic nutritional knowledge (*n* = 4709).

Variable	Lower WS *n* (%)	Higher WS *n* (%)	Chi-Square *p*-Value
Overall Health			
Great	795 (61.1)	507 (38.9)	<0.001
Very Good	902 (54.7)	748 (45.3)	
Good	526 (49.3)	540 (50.7)	
Normal	265 (44.7)	328 (55.3)	
Bad	37 (37.8)	61 (62.2)	
Trouble Sleeping			
Yes	1023 (46.4)	1161 (53.2)	<0.001
No	1502 (59.5)	1023 (40.5)	
Risk of Depression			
Yes	196 (35.4)	357 (64.6)	<0.001
No	2329 (56.0)	1827 (44.0)	
Sedentary Lifestyle			
1–2 h	455 (62.4)	274 (37.6)	<0.001
3–4 h	655 (54.0)	557 (46.0)	
5–6 h	623 (52.1)	572 (47.9)	
More than 6	791 (50.3)	781 (49.7)	
Smoking Cigarette			
Daily Smoker	227 (53.9)	237 (46.1)	0.017
Occasional (Social) Smoker	129 (45.4)	115 (54.6)	
Nonsmoker	2118 (54.2)	1792 (45.8)	
Smoking Waterpipe			
Daily Smoker	96 (55.8)	76 (44.2)	0.005
Occasional (Social) Smoker	184 (45.9)	217 (54.1)	
Nonsmoker	2244 (54.3)	1981 (45.7)	
Nutritional Knowledge Score			
0 out of 4	974 (50.4)	960 (49.6)	0.001
1 out of 4	515 (54.2)	436 (45.8)	
2 out of 4	500 (58.8)	351 (41.2)	
3 out of 4	359 (55.0)	294 (45.0)	
4 out of 4	177 (55.3)	143 (44.7)	

**Table 4 ijerph-18-09141-t004:** Associations between WS and weight management-related variables (*n* = 4709).

Variable	Lower WS *n* (%)	Higher WS *n* (%)	Chi-Square *p*-Value
BMI Category *			
Underweight	189 (66.8)	94 (33.2)	<0.001
Normal	1331 (67.5)	641 (32.5)	
Overweight	682 (47.7)	749 (52.3)	
Obese	323 (31.6)	700 (68.4)	
Weight Misclassification			
No	1516 (55.5)	1217 (44.5)	0.003
Yes	1009 (51.1)	967 (48.9)	
Current Weight Management Action			
Lose Weight	720 (38.5)	1149 (61.5)	>0.001
Manage the Current Weight	722 (70.1)	308 (29.9)	
Gain Weight	217 (59.3)	149 (40.7)	
Nothing	866 (60.0)	578 (40.0)	
Received Doctor Advice About Your Weight			
Yes	308 (35.4)	563 (64.6)	<0.001
No	2216 (56.8)	1621 (42.2)	
Had Bariatric Surgery			
Yes	77 (28.5)	193(71.5)	<0.001
No	2448 (55.1)	1991 (44.9)	

* BMI: Body mass index.

**Table 5 ijerph-18-09141-t005:** Regression model results showing variables related to WS.

Variable	Df	Sig.	Odds Ratio	95% C.I. for EXP (B)	
				Lower	Upper
Age	1	0.000	0.989	0.984	0.994
Region ^$^	12	0.000			
Jouf	1	0.183	1.247	0.901	1.727
Northern Border *	1	0.000	1.980	1.431	2.739
Tabuk *	1	0.029	1.432	1.037	1.977
Madinah	1	0.145	1.274	0.920	1.764
Qassim	1	0.454	1.132	0.819	1.565
Macca *	1	0.036	1.413	1.022	1.954
Riyadh *	1	0.042	1.400	1.013	1.936
Eastern Region	1	0.086	1.331	0.961	1.845
Baha *	1	0.047	1.391	1.005	1.925
Asir *	1	0.001	1.695	1.227	2.342
Jazan	1	0.163	1.263	0.910	1.754
Najran *	1	0.000	2.119	1.529	2.938
Overall Health (with Bad as the reference category)	4	0.004			
Great	1	0.200	0.732	0.454	1.180
Very Good	1	0.446	0.833	0.521	1.332
Good	1	0.760	0.929	0.579	1.490
Normal	1	0.616	1.131	0.699	1.833
Trouble Sleeping (Yes) *	1	0.000	1.371	1.199	1.567
Sedentary Lifestyle (with 1–2 h as the reference category)	3	0.000			
3–4 h *	1	0.000	1.511	1.229	1.858
5–6 h *	1	0.000	1.500	1.218	1.848
More than 6 *	1	0.000	1.527	1.246	1.871
Smoking Waterpipe (with Nonsmoker as the reference category)	2	0.013			
Daily Smoker *	1	0.004	1.803	1.208	2.691
Occasional (Social) Smoker *	1	0.043	1.429	1.012	2.018
Nutritional Knowledge Score (with 0 out of 4 as the reference category)	4	0.000			
1 out of 4 *	1	0.005	0.782	0.659	0.929
2 out of 4 *	1	0.000	0.614	0.511	0.737
3 out of 4 *	1	0.042	0.813	0.665	0.993
4 out of 4 *	1	0.021	0.734	0.563	0.955
BMI Categories (with Underweight as the reference category)	3	0.000			
Normal	1	0.194	1.211	0.907	1.618
Overweight *	1	0.000	2.372	1.740	3.232
Obese *	1	0.000	3.930	2.837	5.444
Current Weight Management Action (with Nothing as the reference category)	3	0.000			
Lose Weight *	1	0.000	1.756	1.496	2.062
Manage the Current Weight *	1	0.002	0.747	0.621	0.897
Gain Weight	1	0.161	1.205	0.928	1.565
Doctor Advice About Your Weight (Yes) *	1	0.000	1.553	1.305	1.849
Had Bariatric Surgery (Yes) *	1	0.000	2.075	1.529	2.815
Risk of Depression (At Risk) *	1	0.000	1.683	1.358	2.087

* Significant *p* value; $ Hail was used as a reference because it was the region with the lowest WS.

## Data Availability

Available from Sharik Association for Health Research upon request.

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
