# Peer review of "Exploring Weight Stigma in Saudi Arabia: A Nationwide Cross-Sectional Study"

_ijerph, 2021, doi:10.3390/ijerph18179141_

Round 1

Reviewer 1 Report

In this study, the authors set out to explore the prevalence of Weight Stigma (WS) and to determine factors that may be associated with it among adults in Saudi Arabia. The authors have provided background about WS, primarily in the US, as it is a topic that hasn’t been explored much in Saudi Arabia. That said, additional information related to this topic and the survey used to complete the study are needed.

Introduction

  • There is a difference between internal weight stigma and external stigma. This paper reads that the focus is external but the survey based on self-perception. The introduction needs to be reframed to address this.
  • Additional background regarding why weight bias is important, particularly in Saudi Arabia, is needed. Not every culture/community looks at overweight/obesity as a negative. Again, the “negatives” of weight are discussed, but they are all from an American perspective. How is weight/overweight perceived in this country in particular?
  • In line 59, it says “…curbing WS is evident.” Again, from the American perspective, it is evident. Provide evidence from the Saudi Arabian perspective (or even another country/culture besides the US).

Methods

Survey

  • The WSSQ needs to be explained – you describe the other variables being measured, but not your main variable.
  • Why was median score chosen (i.e. vs. mean)? Explain.

Discussion

  • Pg 12 – add a heading for WS and nutrition knowledge
  • Pg 13 – line 288-290. This sentence doesn’t make sense/needs revision

There are several typos throughout the paper that need correcting as well.

Author Response

Reviewer #1:

In this study, the authors set out to explore the prevalence of Weight Stigma (WS) and to determine factors that may be associated with it among adults in Saudi Arabia. The authors have provided background about WS, primarily in the US, as it is a topic that hasn’t been explored much in Saudi Arabia. That said, additional information related to this topic and the survey used to complete the study are needed

Authors’ Response: Noted with many thanks for your comments and review.

1- There is a difference between internal weight stigma and external stigma. This paper reads that the focus is external, but the survey based on self-perception. The introduction needs to be reframed to address this.

Authors’ Response: Agree, the introduction been edited and reframed to address the internal weight stigma.

2- Additional background regarding why weight bias is important, particularly in Saudi Arabia, is needed. Not every culture/community looks at overweight/obesity as a negative. Again, the “negatives” of weight are discussed, but they are all from an American perspective. How is weight/overweight perceived in this country in particular?

Authors’ Response: Agree, we included more information about situations in Saudi Arabia or other countries from the same region.

3- In line 59, it says “…curbing WS is evident.” Again, from the American perspective, it is evident. Provide evidence from the Saudi Arabian perspective (or even another country/culture besides the US).

Authors’ Response: Agree, more of the available information related to Saudi Arabia were included.

4- The WSSQ needs to be explained – you describe the other variables being measured, but not your main variable.

Authors’ Response: Agree, more information about WSSQ were included in the method section.

5- Why was median score chosen (i.e. vs. mean)? Explain.

Authors’ Response: Since there was no cut off point, the median has been chosen over the mean because the variable is not normally distributed. This was mentioned in the statistical analysis section.

7- Pg 12 – add a heading for WS and nutrition knowledge

Authors’ Response: Agree, the section have been updated!

8- Pg 13 – line 288-290. This sentence doesn’t make sense/needs revision

Authors’ Response: Done, the sentence has been clarified!

9- There are several typos throughout the paper that need correcting as well.

Authors’ Response: Thank you we used MDPI English editing service to proofread the final manuscript after this revision.

Thank you for your valuable feedback and comments.

Reviewer 2 Report

Dear Authors,

Congratulations on an excellent article, which was a pleasure to read. 

I believe this article is almost ready for publication, although a few details would be helpful for readers outside Saudi Arabia:

  1. A map of the country and the regions listed in the tables would be helpful.
  2. The populations / demographics of these regions would also be useful in order to consider how the sample might be representative of a generally younger or older population.
  3. A note on the proportion of the population who speak Arabic would be useful. Are there many other language spoken in Saudi Arabia?
  4. Another note on the validation process in Arabic would be helpful. Are there cultural factors involved in the translation?
  5. Some consideration of cultural influences from global media and religion might be considered in the discussion. How might the results reflect Saudi Arabia in the 21st century?

Many thanks,

Author Response

Reviewer #2:

Congratulations on an excellent article, which was a pleasure to read. I believe this article is almost ready for publication, although a few details would be helpful for readers outside Saudi Arabia

Authors’ Response: we appreciated your review, many thanks for review this work.

1- A map of the country and the regions listed in the tables would be helpful.

Authors’ Response:. Figure 1 is now included to address this point.

2- The populations / demographics of these regions would also be useful in order to consider how the sample might be representative of a generally younger or older population

Authors’ Response: As mentioned in the analysis section, the final analysis was weighted to account for region size. In figure 1 we included region weight.  

3- A note on the proportion of the population who speak Arabic would be useful. Are there many other language spoken in Saudi Arabia?

Authors’ Response: Agree, the methodology has been updated!

4- Another note on the validation process in Arabic would be helpful. Are there cultural factors involved in the translation?

Authors’ Response: The translated tool used local focus groups to customize the questions. More information about the translated tool were included in methodology section.

5-  Table 1 lacks a breakdown by gender in age groups and regions. This will give a better picture, for example, for comparisons with previous studies in these areas.

Authors’ Response: Done, the table were updated as requested.

6- Some consideration of cultural influences from global media and religion might be considered in the discussion. How might the results reflect Saudi Arabia in the 21st century?

Authors’ Response: Agree, the discussion were updated.

Thank you for your valuable feedback and comments.

Round 2

Reviewer 1 Report

Thank you for your thorough and thoughtful updates and response. This study and its significance to the population involved as well as beyond is much more evident.

One minor correction is sought - In line 94, delete the word "article". 

Thank you and best of luck in your future work.